# Approaches towards Elucidating the Metabolic Program of Hematopoietic Stem/Progenitor Cells

**DOI:** 10.3390/cells11203189

**Published:** 2022-10-11

**Authors:** Hiroshi Kobayashi, Shintaro Watanuki, Keiyo Takubo

**Affiliations:** Department of Stem Cell Biology, Research Institute, National Center for Global Health and Medicine, Tokyo 162-8655, Japan

**Keywords:** stem cell metabolism, metabolome analysis, tracer analysis, single cell metabolic analysis, hematopoietic stem cells, hematopoietic progenitor cells, bone marrow environment

## Abstract

Hematopoietic stem cells (HSCs) in bone marrow continuously supply a large number of blood cells throughout life in collaboration with hematopoietic progenitor cells (HPCs). HSCs and HPCs are thought to regulate and utilize intracellular metabolic programs to obtain metabolites, such as adenosine triphosphate (ATP), which is necessary for various cellular functions. Metabolites not only provide stem/progenitor cells with nutrients for ATP and building block generation but are also utilized for protein modification and epigenetic regulation to maintain cellular characteristics. In recent years, the metabolic programs of tissue stem/progenitor cells and their underlying molecular mechanisms have been elucidated using a variety of metabolic analysis methods. In this review, we first present the advantages and disadvantages of the current approaches applicable to the metabolic analysis of tissue stem/progenitor cells, including HSCs and HPCs. In the second half, we discuss the characteristics and regulatory mechanisms of HSC metabolism, including the decoupling of ATP production by glycolysis and mitochondria. These technologies and findings have the potential to advance stem cell biology and engineering from a metabolic perspective and to establish therapeutic approaches.

## 1. Introduction: Metabolic Programs Regulating Stem Cell Identity and Function

All cells utilize the energy currency adenosine triphosphate (ATP) through the metabolism of nutrients to perform various cellular activities [1,2,3]. Metabolites also maintain intracellular biochemical homeostasis through anabolism and catabolism. Metabolites are the source of proteins and cell membranes, the building blocks of cells, and also support cellular resilience. Several known metabolites regulate the activity of proteins, including various enzymes, and consequently, transcription control, epigenetic regulation, and post-translational modification are also tuned by metabolites [4]. Thus, it is essential to analyze the metabolite levels within cells and the cellular metabolic properties in order to gain a deeper understanding of cellular activities and fates. This is not only true for differentiated cells but also for stem cells with undifferentiated phenotype and self-renewal and multi-differentiation potential.

Tissue homeostasis in living organisms relies on the production and consumption of cells. In other words, the balance between the proliferative capacity and longevity of cells and their replenishment by newly produced cells characterizes each tissue [5]. Among the adult organs, cell turnover is limited in the heart and brain, and the continued use of various types of cells while maintaining their quality over a long period of time is considered to be the essence of tissue homeostasis. On the other hand, the hematopoietic (skin) and gastrointestinal epithelium systems have intense turnover in which differentiated cells are continuously produced and consumed. In these tissues, a continuous supply of differentiated cells is maintained by tissue stem cells that are capable of self-renewal and multilineage differentiation, thereby maintaining tissue homeostasis. Tissue stem cells maintain cell survival and function in the surrounding microenvironment (called the stem cell niche) [6,7]. Cell fate decisions such as symmetric and asymmetric division and differentiation into specific cell types are also thought to be regulated by the stem cell niche. More recently, it has been shown that proper regulation of intracellular metabolism regulated by intrinsic and extrinsic signals is critical for stem cells to remain undifferentiated or to perform various functions [8,9]. These metabolic controls (metabolic programs) that are necessary for stem cells provide energy for cellular functions and maintain the structure of cell membranes and intracellular organelles. The metabolic program not only provides energy currency, such as adenosine-triphosphate (ATP), but also regulates cellular properties themselves. For example, S-adenosylmethionine provides a methyl group, and acetyl CoA provides an acetyl group for protein modifications, including histones. The metabolic program of stem cells is also regulated by environmental factors that constitute the stem cell niche. Furthermore, it is becoming clear that tissue stem cells have context-specific metabolic programs that contribute to the performance of stem cell functions.

In this review, we will first introduce the methodologies and techniques used in the analysis of stem cell metabolic regulation. While these include general approaches used in other fields, this review will also present their advantages over other methods, as well as caveats in adapting them to small numbers of stem cell studies. Next, we will introduce the metabolic characteristics of HSCs, and discuss their significance in stemness.

## 2. Approaches to Metabolic Analysis of Stem Cells (Summarized in Table 1)

### 2.1. Deletion, Knockdown or Overexpression of Specific Genes

Since the introduction of the classical genetic approaches to life sciences, the functional analysis of specific genes has been conducted by various methods. In stem cell metabolism studies, the role of specific genes in the regulation of metabolism in tissue stem cells has been elucidated through the loss, repression, or overexpression of gene function [10,11]. The popular methods are the disruption of specific genes in individual mice by gene targeting or editing techniques such as homologous recombination and CRISPR/Cas9 and the knockdown of gene expression by shRNA and other methods. These loss-of-function gene function studies have led to the identification of not only metabolism-related enzymes but also those genes involved in the regulation of metabolic genes. The use of classical transgenic mice and overexpression vectors such as retroviruses are also important methods. In particular, the use of such vectors to induce gene expression is useful for verifying that the traits observed in knockout mice can be explained by specific metabolic enzymes. For example, consider the possibility that the loss of a gene X reduces the expression of a specific metabolic enzyme gene, which in turn affects stem cell function. In this case, if both stem cell function and the metabolic defect are restored by the overexpression of the reduced expression of a metabolic enzyme gene (introduced by a vector or other means into stem cells lacking that gene X), then that metabolic enzyme gene can be considered the functional target of gene X.

**Table 1 cells-11-03189-t001:** Description, advantages, and disadvantages of the methods used in the approaches to studying stem cell metabolism.

Name	Method	Advantage(s)	Disadvantages(s)
Deletion, knockdown or overexpression of specific genes	Deletion of gene function by gene targeting or editing; knockdown by vectors or oligonucleotides.	Clearly identifies the metabolic effects of each gene on stem cells.	Without spatiotemporally controlled genetic manipulation, only secondary metabolic effects on stem cells may be observed.
Classical metabolite assays using enzymatic reactions	Metabolite detection utilizing enzymatic reactions.	The principle is simple and easy to use. Many kits are commercially available.	Detection sensitivity may be low for use with small numbers of stem cells. There may be no simple enzymatic assay for unusual metabolites.
Mass spectrometry (MS)	Metabolites in intracellular extracts or extracellular fluid are separated by chromatography and measured with a mass spectrometer.	More sensitive than NMR and can identify many metabolites. Levels of unknown metabolites can be identified. Both fate tracking and spatial analysis of metabolites are possible.	Because it is a destructive test, time variation of metabolite levels in the same cell/tissue is unknown. Single cell analysis is under development.
Nuclear magnetic resonance (NMR) spectroscopy	Metabolites in a sample are identified by applying radio waves to the nuclei of molecules placed in a strong magnetic field, causing them to resonate.	Allows for nondestructive metabolite analysis. The fate of atoms derived from metabolite tracers can be accurately determined from which position they were incorporated.	Not sensitive enough to analyze a small number of stem cells. Untargeted analysis of metabolites is difficult.
Chemical probes	Cells take up chemical probes that interact with a metabolite and detect the changes in fluorescence intensity or fluorescence ratio.	A small number of cells can be analyzed. Combination of surface markers enables analysis of stem cells from many cells. Can analyze changes over time in a single cell as well as differences between cells.	Stem cells possess the ability to pump chemical probes out. Even if qualitative analysis is possible, absolute quantification can be difficult.
Genetic biosensors	Genetically encoded fluorescent biosensors to analyze metabolite levels and metabolic characteristics.	A small number of cells can be analyzed. Combination of surface markers enables analysis of stem cells from many cells. Can analyze changes over time in a single cell as well as differences between cells.	Need to select biosensors suitable for the number of metabolites in stem cells. Biosensors must be well expressed in stem cells. Ratiometric sensors are more quantitative.
Flux analyzer	Electrically measure oxygen consumption rate (OCR) and extracellular acidification rate (ECAR) of cells in small wells.	The metabolic state of tens of cells at steady state or under stress load can be measured by adding inhibitors or metabolites over time.	At least tens of thousands of cells are required to obtain an average value that ignores the individuality of each cell. Note that this is a measurement in an in vitro medium and to what extent it reflects OCR and ECAR in the body.

Several points should be noted when explaining the metabolic regulation of stem cells from the deletion or overexpression experiments of the gene of interest. For example, it is possible that the gene itself does not directly regulate stem cell metabolism, but that metabolism is altered in the stem cell secondary to the gene deletion of telomere regulators Terc or Tert [12]. This is due to the fact that metabolic pathways are highly plastic and can compensate for the use of different metabolic pathways rather than using those that are no longer available [13]. It is also difficult to rule out that changes in the number or nature of stem cells or differentiated cells, or alterations in vascular distribution, may alter the consumption of metabolites in the extracellular space or their supply from the cell, resulting in alteration in the function of the stem cell. To avoid these pitfalls, it is necessary to combine spatiotemporally-controlled gene deletion experiments and in vitro analyses to directly estimate the metabolic effects of specific genes in stem cells.

### 2.2. Classical Metabolite Assays Using Enzymatic Reactions

A simple method for quantifying the number of metabolites within cells or in extracellular fluids or serum is to measure the number of metabolites using enzymatic reactions. One easily accessible method is to react the metabolite with an enzyme that uses the metabolite to be measured as (one of) the substrate under the optimal buffer and compound conditions and measure the amount of metabolite produced using a colorimetric method or fluorescence method in a microplate reader. For the quantification of ATP, cell lysate or extracellular fluid is served for measurement by an enzymatic method, while luminescence is determined by a luminometer using the fact that luciferin and ATP are catalyzed by luciferase, ultimately producing oxyluciferin and photons [14]. In this case, a calibration curve is prepared using the metabolite to be measured in the standard product so that the amount of the metabolite in the target sample can be quantified from the colorimetric, fluorescence, and luminescence data obtained. For some representative metabolites, commercially available kits utilizing these principles are available and can be approached relatively easily when conducting research.

Even in classical metabolite assays using these enzymatic reactions, there are caveats to be aware of; for example, there are sometimes only a few highly-sensitive assays available for small numbers of cells, such as tissue stem cells. Also, in commercially available kits, the concentration ranges of the standards for manually setting calibration curves are often not optimized for metabolite analysis based on small numbers of cells. Therefore, it is important to prepare calibration curves optimized using reference materials, including the low-concentration range.

### 2.3. Metabolite Analysis by Mass Spectrometry (MS) and Nuclear Magnetic Resonance (NMR) Spectroscopy

A more direct approach than the indirect measurement of metabolites using enzymatic reactions is the measurement of metabolites by MS or NMR. Among the omics analyses, the methodology to comprehensively measure metabolites using MS [15,16,17], NMR [18], etc., is called metabolomic analysis. MS is more sensitive than NMR and is particularly useful in stem cell research where the number of cells is limited. Hundreds to thousands of metabolites can be analyzed from a single sample. In particular, metabolites can be separated and identified with high accuracy if samples are separated by various types of chromatography prior to the analysis by MS [15,16,17]. In addition, MS analysis does not necessarily require narrowing down the metabolites to be measured prior to analysis, making it possible to search for uncharacterized metabolites. Also, by adding isotope-labeled metabolite tracers to stem cells, the fate of the metabolites incorporated in the stem cells can be analyzed. On the other hand, NMR-based metabolomic analysis is often less sensitive than most MS-based analyses, but it can be performed noninvasively in vivo. In fact, in vivo applications of NMR, such as MRI, can be used to analyze the distribution and spatio-temporal changes of metabolites in vivo [19]. In addition, in the tracking of isotope tracers, it is possible to accurately determine the position of the labeled isotope in the metabolite.

Each of these methods has its advantages and is very powerful in the analysis of metabolites in stem cells. However, although techniques for the analysis of small numbers of cells have been developed [20], sufficient numbers of cells are still required, and it is not easy to analyze which of the metabolites found by non-targeted metabolomics analysis are important metabolites. In addition, single-cell analysis, which has been advanced in other omics analyses, such as transcriptome, is still technically under development compared to bulk analysis [21], and further technological breakthroughs are expected in the near future.

### 2.4. Measurement Using Chemical Probes and Genetic Biosensors

The measurement of metabolites using enzyme reactions or MS is a destructive test, and the cells and tissues to be analyzed are destroyed to extract the metabolites. Therefore, the changes in the metabolites over time in the same cell remain unknown. Also, even when using the most sensitive method currently available that can detect a sufficient number of metabolites, several thousand cells are required for the enzymatic reactions and MS measurements. This is a problem when conducting metabolic research with tissue stem cells, which have only a limited number of cells in the body. There are several known approaches for analyzing the metabolic properties of cells and the intracellular metabolites in living cells that resolve these problems.

The first method is to use chemical probes that are taken up by cells or penetrate the cell membranes. For example, in examining the uptake of glucose, a fluorescent derivative of glucose, 2-NBDG [22,23], is useful. Aldehyde dehydrogenase (ALDH) is known to be highly active in several stem cell types [24,25,26]. It can be used to identify changes in ALDH activity in stem cells and to preliminarily identify stem cell fractions using a fluorescent probe for the ALDH activity as an indicator. Another example is reduced glutathione (GSH). Normal and malignant stem cells are known to have high intracellular levels of GSH and are known to be highly resistant to oxidative stress. The use of fluorescent chemical probes for GSH (e.g., QuicGSH3.0 or SCT 036 BioTracker™ 625 Red GSH Dye) [27] that can measure the amount of GSH will be useful for the evaluation and mechanistic analysis of the antioxidant capacity of tissue stem cells and the development of targeted therapies against cancer stem cells.

On the other hand, the use of genetically encoded fluorescent biosensors is also useful in stem cell metabolism studies. Two main types of biosensors are utilized for the detection of metabolites. One is a monomeric molecule that fluoresces when it undergoes a conformational change depending on the target metabolite of detection. The other sees two different fluorescent proteins connected by a linker, for which the conformational changes in the metabolite of detectable interest and the Förster resonance energy transfer (FRET) that occurs in response to those changes are detected. For example, there is an example of a biosensor of lactic acid being used to analyze cellular communications [28]. In another example, the fluorescent probe mVenus-TOSI, which detects mTORC1 activity, a highly sensitive indicator of a cell’s metabolic state, is used to examine the stemness of acute myeloid leukemia cells [29]. Furthermore, the metabolic characteristics of hematopoietic stem/progenitor cells (HSPCs) can now be analyzed using knock-in mice with the ATP biosensor, which can measure intracellular ATP concentration [30].

Thus, fluorescent chemical probes and biosensors, in combination with another fluorophore-labeled antibody staining, can identify the metabolic characteristics and metabolite levels of rare cellular fractions such as tissue stem cells in a large cell population. Another advantage is the ability to identify the metabolic characteristics of each cell. This is a unique feature when considering that single-cell metabolome analysis is still technically difficult. For the use of these probes and biosensors, there are specific issues to be considered. When performing quantitative rather than qualitative analysis, a calibration curve should be prepared that fits the concentration range of the metabolite in the stem cells, using standards. Conversely, when interpreting the results of the analysis, it is important to pay attention to whether the number of metabolites in the analyzed cells is compatible with the dynamic range of the probe or sensor itself. On the other hand, it should be noted that because these approaches are nondestructive tests, the correct measurement results cannot be obtained if the probe uptake into the cell is low or if the expression of the biosensor is faint. The selection of optimal chemical probes and the use of ratiometric biosensors can be useful in avoiding these pitfalls. Genetically encoded fluorescent biosensors often have difficulty expressing a sufficient level of either too much or too little in the in vivo transgenic models. Even if they can be successfully generated, the researchers should carefully check the cell-to-cell variation in the expression levels [23].

### 2.5. Flux Analyzer

The flux analyzer is a unique and commercially available instrument. Tens of thousands (or more) of cells can be seeded in wells that resemble cell culture plates, and their oxygen consumption rate (OCR) and extracellular acidification rate (ECAR) can be measured using electrodes in the culture medium. ECAR is thought to reflect lactate production by anaerobic glycolysis. Because this instrument can sequentially add various inhibitors, metabolites, and compounds, it can infer changes in OXPHOS and glycolysis by measuring how cells exhibit OCR and ECAR when subjected to various stresses. For example, metabolic changes in HSCs and HPCs during aging have been reported using a flux analyzer [31]. One point to note is that, when measuring blood cells, such as HSCs, it is necessary to stick the cells to the bottom of the measurement plate.

## 3. HSCs and Metabolic Program

### 3.1. HSCs and Their Metabolic Programs Which Maintain Individual Hematopoiesis

As an example of stem cell metabolism research, applying the various metabolic analysis techniques described above, here we present the findings on HSC and HPC metabolism. Of the cells that make up our bodies, about two-thirds are blood cells, mainly erythrocytes. Blood cells play a variety of roles, such as oxygen transport, immune response, and hemostasis, and are indispensable for sustaining organismal life [32]. Blood cells produced in mammals after birth are derived from HSCs and HPCs that reside in the bone marrow. HSCs are typical tissue stem cells that retain the ability to self-renew and differentiate into all blood cells [7,33]. After differentiating into multipotent progenitors with a reduced self-renewal capacity, HSCs differentiate into more committed HPCs with limited differentiation capacity, such as myeloid and lymphoid progenitors, and finally give rise to terminally differentiated blood cells. HSCs are also used in HSC transplantation, a curative therapy for malignant hematopoietic tumors, in which the recipient’s bone marrow is reconstructed and the hematopoietic system is replaced by blood cells derived from donor HSCs. In the steady state, HSCs are in a quiescent state of the cell cycle (G0 phase), and, together with multipotent progenitor cells, they contribute to the continuous production of blood cells. When acute stresses such as transplantation, hemorrhage, inflammation, or infection are loaded, HSCs leave the G0 state and actively proliferate to provide HPCs and differentiated blood cells, contributing to the restoration of homeostasis in the hematopoietic system [34].

In order for HSCs to function properly both under steady-state conditions and under stress, the molecular mechanisms that define the identity of stem cells, mainly transcription factors, must be activated. In particular, the production and consumption of ATP, the energy currency, and the maintenance of the nuclear genome, epigenetic state, cell membrane, and intracellular organelles by nucleic acids and lipids are essential for various events such as cell survival, division, and differentiation (Figure 1). The elucidation of these metabolic programs in HSCs and the development of HSC manipulation techniques based on these programs are important research trends [35].

### 3.2. Metabolic Control of HSPCs in Steady State

#### 3.2.1. Hypoxic Microenvironment and HSCs

Oxygen is required for ATP production in mitochondria. Thus, cellular energy production and oxygen demand are inseparable. While the bone marrow has a high oxygen demand for blood cell production, oxygen supply is dependent on perfusion by blood vessels that penetrate the bone and enter the marrow cavity. In other words, it is difficult to sufficiently and timely increase the oxygen supply through angiogenesis to meet the demand for oxygen and nutrients in the bone marrow. For this reason, bone marrow has long been considered a hypoxic environment. Using the hypoxia probe pimonidazole, HSCs were found to be even more hypoxic than other differentiated blood cells [36,37,38]. Recently, oxygen partial pressure in bone marrow was measured using a combination of in vivo imaging and phosphorescent probes. The analysis reported that the bone marrow is an extremely hypoxic organ with an oxygen partial pressure of approximately 10 mmHg [39,40]. Indeed, the activation of the hypoxia sensor transcription factor hypoxia-inducible factor-1α (HIF-1α) is observed in HSCs. HSCs lacking HIF-1α were functionally impaired as a result of increased cell cycling, and their stem cell activity decreased [36]. This was thought to be due in part to the inability of HSCs to maintain their expression of GRP78, the receptor for Cripto, which maintains stem cell activity [41]. In addition, HIF-1α deficiency causes HSCs to leave the bone marrow niche and emerge in the periphery [36]. HIF-1α acts as one of the hubs of HSC regulation; for example, HIF-1α reduction due to transcription factor ID2 deficiency leads to HSC activation and depletion [42]. Overall, the hypoxic environment in the bone marrow is an environmental factor that could contribute to the maintenance of HSCs.

#### 3.2.2. Regulation of Glycolysis and Decoupling from TCA Cycle in HSCs

Metabolomic analysis using capillary electrophoresis time-of-flight MS (CE-TOFMS) suggested that the activity of phosphofructokinase (PFK), the rate-limiting enzyme of glycolysis, is high in HSCs, and that the metabolite flux from glycolysis to mitochondria is suppressed [23]. HIF-1α is also known as a regulator of glycolysis. In fact, HIF-1α induces the gene expression of various glycolytic enzymes in HSCs. Among the glycolytic enzymes, the deficiency of LDHA, a subunit of the lactate dehydrogenase complex, strongly impairs the number and function of hematopoietic stem cells, suggesting the importance of the glycolytic system for the maintenance of HSCs [43]. Interestingly, in mice lacking PKM2, an isozyme of pyruvate kinase and also an enzyme of glycolysis, HSCs are functionally normal, but HPCs are impaired [43]. These observations suggest that the metabolic program of HSPCs is maintained at least in part by the expression of specific metabolic enzymes themselves. HIF-1α also induces the expression of pyruvate dehydrogenase kinase (Pdk), which phosphorylate and inactivate pyruvate dehydrogenase (PDH). Among the four Pdk family members, HIF-1α-deficient HSCs show reduced expression of Pdk2 and Pdk4 [23]. PDH catalyzes the reaction that converts pyruvate to acetyl CoA, which has the effect of driving the mitochondrial tricarboxylic acid (TCA) cycle. Pdk functions as a negative regulator of the metabolic flux from glycolysis to the mitochondrial TCA cycle. Thus, HIF-1α in HSCs suppresses the metabolic flux into the TCA cycle from glycolysis by Pdk expression. This indicates that ATP production by glycolysis, which does not require oxygen, and mitochondrial ATP production by oxidative phosphorylation (OXPHOS), which does require oxygen, are decoupled. Rather than having the glycolysis and mitochondrial OXPHOS integrated into one ATP-producing system through aerobic glycolysis, the independent existence of the two ATP-producing pathways (anaerobic glycolysis and OXPHOS) may provide stress tolerance for the HSCs. In fact, HSCs lacking these two Pdk family genes showed not only altered metabolic characteristics, such as a decrease in glycolysis and increased mitochondrial mass, but also the activation of cell cycling, accumulation of oxidative stress, and decreased stem cell activity [23]. Thus, the metabolic properties of HSCs, which maintain the decoupling of glycolytic and mitochondrial ATP production via the HIF-1α-Pdk axis, are necessary for maintaining the phenotypes and functions of HSCs.

#### 3.2.3. Mitochondrial Dynamics and Steady-State HSCs

Mitochondrial activity and quantity fission and fusion alter the number and functions of HSCs in a steady state or during stress. Based on analysis using transmission electron microscopy and mitochondrial probes, the number of mitochondria in HSCs was thought to be low. However, it has been reported that HSCs have the ability to pump mitochondrial probes out of the cell and that their actual mitochondrial mass is equal to or rather higher than that of differentiated cells [44]. Interestingly, flux analyzer analysis has shown that mitochondrial oxygen consumption is suppressively regulated in HSCs [44]. These findings, together with the fact that metabolic flux from glycolysis to the TCA cycle is suppressed under a steady state, suggest that mitochondrial activity may be suppressed in HSCs. However, this does not mean that mitochondria themselves are unnecessary at all. For example, the mitochondrial enzyme PTPMT1 (PTEN-like phosphatase) is required for the maintenance of mitochondria in HSCs and their differentiation potential [45]. In addition, the loss of fumarate hydratase (Fh1), an enzyme of the TCA cycle, impairs the self-renewal and differentiation potential of HSCs [46]. Furthermore, the loss of Mitofusin 2 (Mfn2), which regulates mitochondrial fusion and division, maintains the ability of HSCs to differentiate into myeloid cells, but impairs differentiation into lymphoid lineages [47]. More directly, the loss of the complex III subunit Rieske iron-sulfur protein (RISP) of mitochondrial OXPHOS leads to the proliferation of HSCs along with HPCs, impairing the quiescence, differentiation potential, and survival of fetal and adult HSCs [48].

#### 3.2.4. Environmentally-Derived Metabolites That Maintain HSC Homeostasis

Not only the environment near the HSCs but also the metabolites derived from the diet, are important for the metabolic control of steady-state HSCs. Vitamin A supplied by food activate retinoic acid (RA) signaling functions to maintain quiescence by inhibiting reactive oxygen species production and protein translation in HSCs [49]. Importantly, non-classical RA signaling by the Cyp26b1-mediated production of 4-oxo-RA regulates HSC function via RA receptor beta [50]. Another example is ascorbate/vitamin C. Metabolomic profiling showed that human and mouse HSCs had unusually high levels of ascorbate, which decreased upon differentiation. The systemic depletion of ascorbate in mice increased HSC frequency and function, in part by the decreased function of Tet2, a dioxygenase tumor suppressor. Ascorbate depletion accelerated leukemogenesis induced by an oncogenic mutant of Flt3. Thus, the accumulation of ascorbate in HSCs increases Tet activity to limit HSC frequency and suppress leukemogenesis [51].

### 3.3. Metabolic Regulation of HSCs under Stress

#### 3.3.1. Acute Stress and HSC Metabolism

HSCs are not always quiescent in the cell cycle. Various stresses, or the loss of differentiated blood cells (infection, hemorrhage, transplantation, etc.), activate the HSCs to proliferate symmetrically or asymmetrically and provide differentiated cells. This is a fundamental activity of the HSC to maintain homeostasis in the blood system. Various stress loads activate energy production by mitochondria in HSCs. For example, fatty acid β-oxidation induced by PPARδ reportedly supports HSC self-renewal divisions. Stress-induced mitochondrial activation increases ROS as a byproduct. As a result, the stress sensor p38MAP kinase is activated. HSCs deficient in p38α, the major p38MAP kinase isozyme in blood cells, have impaired hematopoietic recovery with decreased cell cycling after stress loading [52]. In the stressed HSCs, p38α initiates CREB-Mitf-inosine monophosphate dehydrogenase 2 (Impdh2) signaling to activate purine metabolism. Impdh2 is the rate-limiting enzyme in the biosynthetic pathway for guanine nucleotide, following the pentose phosphate pathway, a branch pathway of glycolysis. This was found in metabolite profiling via the CE-TOFMS of HSCs after stress loading, as amino acids that should enter the purine biosynthetic pathway accumulate in p38α deficiency. Additionally, HSPCs show a strong dependence on hypoxanthine guanine phosphoribosyl transferase (HPRT)-associated purine salvaging. HPRT deficiency resulted in altered cell-cycle progression, proliferation kinetics, and mitochondrial membrane potential in HSCs, whereas the HPCs were less affected [53]. Thus, the activation of purine metabolism is critical for HSC function upon stress. However, it remains unclear how the overall central carbon metabolism is readjusted during stress loading.

#### 3.3.2. Mitochondrial Activity and Acutely Stressed HSCs

The mitochondrial activation of HSCs by proliferative stress is induced by elevated Ca2+ levels in the cytoplasm [54]. This is suppressed under a steady state by the adenosine supplied by the neighboring cells. Based on these results, HSC mitochondria are an important regulator of HSC numbers and function. The electron transport chain in mitochondria supports aspartate production. Aspartate levels in HSCs and HPCs are dependent on intracellular synthesis and increase upon HSC activation. The regulation of aspartate levels is well-coupled with HSC function [55]. The overexpression of the glutamate/aspartate transporter, Glast, or the loss of the glutamic-oxaloacetic, transaminase 1 (Got1), each increase the aspartate levels in HSPCs and increase the functioning in HSCs but not HPCs. In contrast, the loss of Got2 reduces aspartate and the functioning in HSCs but not HPCs. The deletion of both Got1 and Got2 eliminated the HSCs. By using isotope tracing, aspartate was identified as a source for the synthesis of asparagine and purines [55]. Both contributed to increased HSC function during hematopoietic regeneration. During proliferative stress, metabolism also regulates the epigenetic status of the HSCs. The acetyl CoA supply via ATP citrate lyase supports differentiated cell production via increased histone acetylation in HSCs [56].

#### 3.3.3. Aging Stress and HSC Metabolism

The quality control of mitochondria and other intracellular organelles is important in the metabolic regulation of old stem cells, including HSCs. It is known that mitochondrial accumulation and metabolic activation due to decreased autophagy during aging leads to decreased HSC function [31]. Interestingly, one-third of old HSCs still perform autophagy properly, contributing to the maintenance of hematopoiesis through the healthy HSCs after aging [31]. Recently, another intracellular organelle quality control mechanism, chaperone-mediated autophagy (CMA), was found to be associated with HSC aging. CMA is required for fatty acid metabolism upon HSC activation and CMA activity in the HSCs decreases with age [57]. The restoration of CMA in old HSCs can restore the functionality of old mouse and human HSCs [57]. Apart from mitochondrial mass, an appropriate level of mitochondrial membrane potential (MMP) is also an indicator of the quality of HSCs as they age [58]. In this study, MMPs in HSCs are measured using TMRM, a fluorescent MMP probe; the addition of TMRM can be reliably measured by an inhibitor of the ABC transporter (verapamil). The pharmacological restoration of MMP improves the function of old HSCs [58]. Other metabolic pathways besides mitochondria and HSC aging are still unclear. In HSCs, p38α inhibits aging phenotypes in the early aging stage but promotes aging phenotypes in the late aging stage, and purine metabolism may also undergo a time-dependent change [59]. These reports suggest that the reactivation of young HSC-like metabolism may be a promising approach to rejuvenating HSC after aging.

### 3.4. Application of Metabolic Findings of HSC

#### 3.4.1. Metabolic Optimization of HSC Culture

Attempts are also being made to apply the knowledge gained from HSC metabolism research to stem cell culture or manipulation. For example, the treatment of HSCs with the RA signaling agonist all-trans retinoic acid (ATRA) was shown to maintain the immature phenotype, low biosynthetic activity, and cell cycle quiescence [49]. ATRA-treated HSCs showed an improved serial plating capacity in vitro and performed with a better serial reconstitution capacity in vivo compared to control cells. Another example is the use of a pyruvate analog, 1-aminoethylphosphinic acid (1-AA), that mimics the action of Pdk. The treatment of HSCs with 1-AA maintained the metabolic program and cellular quiescence of HSCs in vitro in the bone marrow. As a result, HSCs can be cultured for long periods of time while retaining their transplantation activity [23]. 

Recently, metabolites in the bone marrow have been shown to constitute a niche of environmental factors that regulate HSCs. Since adipocyte is known to increase in bone marrow, especially with aging, a close relationship between HSCs and fatty acids in bone marrow was assumed. Outside the cell, fatty acids are mainly bound to albumin. In HSC cultures, only a much lower concentration of albumin (0.1–1%) is added to the culture than the blood concentration of albumin (about 4%). In fact, when HSCs are cultured in vitro in hypoxic conditions with the addition of sufficient amounts of fatty acid-bound albumin, they maintain cell cycle quiescence and transplant survival [60]. Although the mode of fatty acid supply from intramedullary fat to the HSCs remains unclear, in vivo imaging and lipidomics analyses suggest that fatty acids abundantly loaded with circulating albumin may diffuse deep into the bone marrow HSC niche to supply the fatty acids.

#### 3.4.2. Non-Conditional Transplantation with the Removal of Metabolites from the Diet

Another environmentally-derived metabolite for HSCs is valine. By applying MS technology to tissue sections, one amino acid (valine) has been revealed to be abundantly distributed in bone marrow [61]. In fact, when amino acids were removed one by one from the culture medium and examined, the removal of valine, in particular, resulted in a failure to maintain the HSCs. Also, mice fed a valine-deficient diet become recipients that can accept HSC transplantation without pretreatment as a result of the removal of the HSC cells from the bone marrow niche [61].

## 4. Conclusions and Perspective

Understanding the biological properties and regulatory mechanisms of stem cells from a metabolic perspective is an important issue that will contribute to the development of fundamental technologies for regenerative medicine. In recent years, various metabolite profiling and cellular metabolic trait analysis methods described in this review have revealed the existence and significance of the metabolic program in stem cells. The studies into the metabolic program used in HSCs presented here have led to this trend, showing that metabolic regulation is deeply involved in the maintenance and dynamics of HSCs in the bone marrow. In the future, metabolome analysis using a small number of cells (hopefully single cells) and isotope-labeled flux analysis will become increasingly important in this field. In addition, MS imaging technology with high spatial resolution information is also a highly needed technological platform to provide metabolic information on cellular communities in niche areas. There have already been reports of attempts to analyze the metabolism of a small number of cells by in vivo tracing [55,62]. These future technologies will allow us to understand the metabolic changes and mechanisms during the differentiation and proliferation of HSCs at the single-cell level with high spatiotemporal resolution and to study the metabolic dynamics during HSC fate determination. The close collaboration between stem cell biology and the development of metabolic analysis methods will lead tissue stem cell research, including HSCs, to more depth and breadth. Furthermore, the clinical application of these metabolic findings will lead to therapeutic strategies for hematological tumors derived from HSCs and HPCs, as well as to the amplification and functional enhancement of HSCs through metabolic modulation, thereby advancing tumor treatment and regenerative medicine.

## Figures and Tables

**Figure 1 cells-11-03189-f001:**
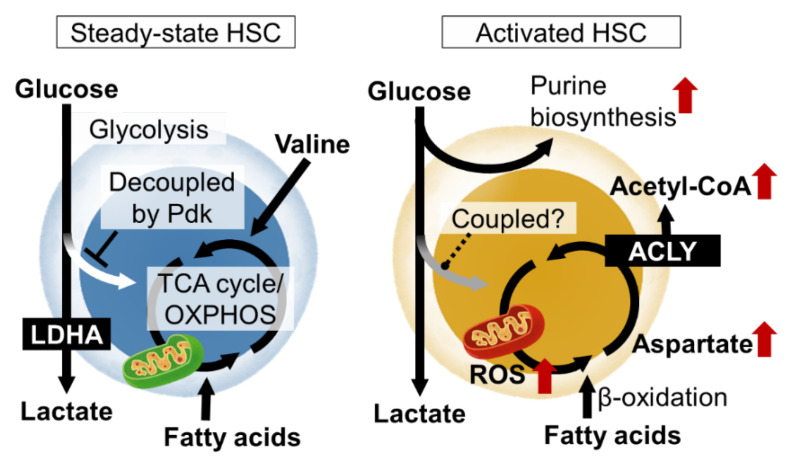
Nutrient requirements and metabolic programs of hematopoietic stem cells (HSCs) in steady state and upon activation. In steady-state HSCs, glycolytic and mitochondrial ATP production are decoupled. Various metabolic pathways are activated in HSCs upon stress loading or cytokine stimulation, resulting in metabolic reprogramming that induces proliferation and differentiation. LDHA, lactate dehydrogenase A; ROS, reactive oxygen species; Pdk, pyruvate dehydrogenase kinase; TCA, tricarboxylic acid; OXPHOS, oxidative phosphorylation; ACLY, ATP citrate lyase.

## Data Availability

Not applicable.

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
