# Peer review of "Approaches towards Elucidating the Metabolic Program of Hematopoietic Stem/Progenitor Cells"

_cells, 2022, doi:10.3390/cells11203189_

Round 1

Reviewer 1 Report

The first half of this manuscript is devoted to methodological aspects as they are applied to hematological stem cell research. There does not seem to be something special as regards this use as compared to that with other cell types. As such it appears, on a whole, as unnecessary or, at least, could be greatly shortened. The second part of the manuscript lists a series of observations on the metabolic profile of hematopoietic stem cells and hematopoietic progenitor cells. The observations, however, are put together without a visible common thread. They appear to be merely a descriptive patchwork of results obtained over the years.

Author Response

Responses to Reviewer 1

The first half of this manuscript is devoted to methodological aspects as they are applied to hematological stem cell research. There does not seem to be something special as regards this use as compared to that with other cell types. As such it appears, on a whole, as unnecessary or, at least, could be greatly shortened.

Thank you for your valuable comments. We, too, have carefully considered whether to add these sentences in the original submission, but we included them to make the review useful to hematology and/or stem cell researchers who wish to begin analyzing metabolism. While these are not necessary for experts like you and me who are well understand stem cell metabolism, we believe that this is an important attempt at a comprehensive review that will be useful to the broad spectrum of readers for which this series is intended. Therefore, while we greatly respect your suggestions, we would like to leave these descriptions in the revision. To clarify this point, the intent of this review was added to “1. Introduction” (shown in blue).

The second part of the manuscript lists a series of observations on the metabolic profile of hematopoietic stem cells and hematopoietic progenitor cells. The observations, however, are put together without a visible common thread. They appear to be merely a descriptive patchwork of results obtained over the years.

We appreciate your comments that improve the manuscript. We have tried to add more headings and totally reorganize the latter part (section 3) of the manuscript (shown in blue). The revised manuscript is organized to systematically discuss metabolic changes in HSCs at steady state and during stress, respectively. The metabolic regulation of mitochondria is also included, as well as a section on metabolic changes in aging HSCs as a representative of chronic stress (section 3.3.3. in the revised manuscript).

Reviewer 2 Report

In the manuscript “Approaches towards elucidating the metabolic program of hematopoietic stem/progenitor cells”, the authors proposed to discuss the methodologies and techniques used in the analysis of stem cell metabolic regulation, in addition to the metabolic characteristics of HSCs while discussing their significance in stemness. The paper could be of interest but there are many drawbacks that hampered the initial enthusiasm. The authors are encouraged to include more information and details regarding some topics included on this report.  I have several suggestions that may be useful.

Specific comments:

1. It is essential to include the advantages and discuss the disadvantages of the different approaches used to study the metabolic analysis of stem cell. The information included in table 1 is a good summary but it is crucial to get the authors point to each technical approach. Examples for the specific application of each would also be of good value.

2. figure 1 needs some work. It is difficult to follow the author’s point. The font is small and the image could be larger.

3. There is discussion on the future perspectives on the topic. What fields are expected to grow? Which pathway is research taking? Are there new methodological approaches on the way? In addition, there is no take home message nor discussion on the clinical relevance of studies on metabolic analysis of stem cells.

Author Response

Responses to Reviewer 2

  1. It is essential to include the advantages and discuss the disadvantages of the different approaches used to study the metabolic analysis of stem cell. The information included in table 1 is a good summary but it is crucial to get the authors point to each technical approach. Examples for the specific application of each would also be of good value.

We appreciate your valuable comments that improve the manuscript. To further improve Table 1, a description of the flux analyzer has been added, and a new section has been added with an example of using the flux analyzer (section 2.5). In addition, we included Terc and Tert as examples of molecules that are not directly regulated by knockout mice, and included examples of HSC metabolic analysis using MS and chemical probes (in sections 2.1., 3.3.1., and 3.3.3., shown in blue).

  1. figure 1 needs some work. It is difficult to follow the author’s point. The font is small and the image could be larger.

We have improved the figure by enlarging the font and illustrations as you suggested.

  1. There is discussion on the future perspectives on the topic. What fields are expected to grow? Which pathway is research taking? Are there new methodological approaches on the way? In addition, there is no take home message nor discussion on the clinical relevance of studies on metabolic analysis of stem cells.

We have added to section 4 about future prospects and expectations for contributions to clinical medicine (shown in blue).

Round 2

Reviewer 1 Report

The authors have improved the manuscript according to the referees' demands and I don not ask for further changes

Reviewer 2 Report

The revised version has improved